# Online Learning with Adversarial Delays

Kent Quanrud[*] and Daniel Khashabi[†]

Department of Computer Science
University of Illinois at Urbana-Champaign
Urbana, IL 61801
{quanrud2,khashab2}@illinois.edu

## Abstract

We study the performance of standard online learning algorithms when the feedback is delayed by an adversary. We show that `online-gradient-descent` [1] and `follow-the-perturbed-leader` [2] achieve regret $O(\sqrt{D})$ in the delayed setting, where $D$ is the sum of delays of each round's feedback. This bound collapses to an optimal $O(\sqrt{T})$ bound in the usual setting of no delays (where $D = T$). Our main contribution is to show that standard algorithms for online learning already have simple regret bounds in the most general setting of delayed feedback, making adjustments to the analysis and not to the algorithms themselves. Our results help affirm and clarify the success of recent algorithms in optimization and machine learning that operate in a delayed feedback model.

## 1 Introduction

Consider the following simple game. Let $K$ be a bounded set, such as the unit $\ell_1$ ball or a collection of $n$ experts. Each round $t$, we pick a point $x_t \in K$. An adversary then gives us a cost function $f_t$, and we incur the *loss* $\ell_t = f_t(x_t)$. After $T$ rounds, our *total loss* is the sum $L_T = \sum_{t=1}^{T} \ell_t$, which we want to minimize.

We cannot hope to beat the adversary, so to speak, when the adversary picks the cost function *after* we select our point. There is margin for optimism, however, if rather than evaluate our total loss in absolute terms, we compare our strategy to the best fixed point in hindsight. The *regret* of a strategy $x_1, \ldots, x_T \in K$ is the additive difference $R(T) = \sum_{t=1}^{T} f_t(x_t) - \arg\min_{x \in K} \sum_{t=1}^{T} f_t(x)$.

Surprisingly, one can obtain positive results in terms of regret. Kalai and Vempala showed that a simple and randomized follow-the-leader type algorithm achieves $R(T) = O(\sqrt{T})$ in expectation for linear cost functions [2] (here, the big-$O$ notation assumes that the diameter of $K$ and the $f_t$'s are bounded by constants). If $K$ is convex, then even if the cost vectors are more generally convex cost functions (where we incur losses of the form $\ell_t = f_t(x_t)$, with $f_t$ a convex function), Zinkevich showed that gradient descent achieves regret $R(T) = O(\sqrt{T})$ [1]. There is a large body of theoretical literature about this setting, called *online learning* (see for example the surveys by Blum [3], Shalev-Shwartz [4], and Hazan [5]).

Online learning is general enough to be applied to a diverse family of problems. For example, Kalai and Vempala's algorithm can be applied to online combinatorial problems such as shortest paths [6], decision trees [7], and data structures [8, 2]. In addition to basic machine learning problems with convex loss functions, Zinkevich considers applications to industrial optimization, where the

---

[*]`http://illinois.edu/~quanrud2/`. Supported in part by NSF grants CCF-1217462, CCF-1319376, CCF-1421231, CCF-1526799.

[†]`http://illinois.edu/~khashab2/`. Supported in part by a grant from Google.

value of goods is not known until after the goods are produced. Other examples of applications of online learning include universal portfolios in finance [9] and online topic-ranking for multi-labeled documents [10].

The standard setting assumes that the cost vector $f_t$ (or more generally, the feedback) is given to and processed by the player before making the next decision in round $t + 1$. Philosophically, this is not how decisions are made in real life: we rush through many different things at the same time with no pause for careful consideration, and we may not realize our mistakes for a while. Unsurprisingly, the assumption of immediate feedback is too restrictive for many real applications. In online advertising, online learning algorithms try to predict and serve ads that optimize for clicks [11]. The algorithm learns by observing whether or not an ad is clicked, but in production systems, a massive number of ads are served between the moment an ad is displayed to a user and the moment the user has decided to either click or ignore that ad. In military applications, online learning algorithms are used by radio jammers to identify efficient jamming strategies [12]. After a jammer attempts to disrupt a packet between a transmitter and a receiver, it does not know if the jamming attempt succeeded until an acknowledgement packet is sent by the receiver. In cloud computing, online learning helps devise efficient resource allocation strategies, such as finding the right mix of cheaper (and inconsistent) spot instances and more reliable (and expensive) on-demand instances when renting computers for batch jobs [13]. The learning algorithm does not know how well an allocation strategy worked for a batch job until the batch job has ended, by which time many more batch jobs have already been launched. In finance, online learning algorithms managing portfolios are subject to information and transaction delays from the market, and financial firms invest heavily to minimize these delays.

One strategy to handle delayed feedback is to pool independent copies of a fixed learning algorithm, each of which acts as an undelayed learner over a subsequence of the rounds. Each round is delegated to a single instance from the pool of learners, and the learner is required to wait for and process its feedback before rejoining the pool. If there are no learners available, a new copy is instantiated and added to the pool. The size of the pool is proportional to the maximum number of outstanding delays at any point of decision, and the overall regret is bounded by the sum of regrets of the individual learners. This approach is analyzed for constant delays by Weinberger and Ordentlich [14], and a more sophisticated analysis is given by Joulani *et al.* [15]. If $\alpha$ is the expected maximum number of outstanding feedbacks, then Joulani *et al.* obtain a regret bound on the order of $O(\sqrt{\alpha T})$ (in expectation) for the setting considered here. The blackbox nature of this approach begets simultaneous bounds for other settings such as partial information and stochastic rewards. Although maintaining copies of learners in proportion to the delay may be prohibitively resource intensive, Joulani *et al.* provide a more efficient variant for the stochastic bandit problem, a setting not considered here.

Another line of research is dedicated to scaling gradient descent type algorithms to distributed settings, where asynchronous processors naturally introduce delays in the learning framework. A classic reference in this area is the book of Bertsekas and Tsitskilis [16]. If the data is very sparse, so that input instances and their gradients are somewhat orthogonal, then intuitively we can apply gradients out of order without significant interference across rounds. This idea is explored by Recht *et al.* [17], who analyze and test parallel algorithm on a restricted class of strongly convex loss functions, and by Duchi *et al.* [18] and McMahan and Streeter [19], who design and analyze distributed variants of adaptive gradient descent [20]. Perhaps the most closely related work in this area is by Langford *et al.*, who study the `online-gradient-descent` algorithm of Zinkevich when the delays are bounded by a constant number of rounds [21]. Research in this area has largely moved on from the simplistic models considered here; see [22, 23, 24] for more recent developments.

The impact of delayed feedback in learning algorithms is also explored by Riabko [25] under the framework of "weak teachers".

For the sake of concreteness, we establish the following notation for the delayed setting. For each round $t$, let $d_t \in \mathbb{Z}^+$ be a non-negative integer *delay*. The feedback from round $t$ is delivered at the end of round $t + d_t - 1$, and can be used in round $t + d_t$. In the standard setting with no delays, $d_t = 1$ for all $t$. For each round $t$, let $\mathcal{F}_t = \{u \in [T] : u + d_u - 1 = t\}$ be the set of rounds whose feedback appears at the end of round $t$. We let $D = \sum_{t=1}^{T} d_t$ denote the sum of all delays; in the standard setting with no delays, we have $D = T$.

In this paper, we investigate the implications of delayed feedback when the delays are *adversarial* (i.e., *arbitrary*), with no assumptions or restrictions made on the adversary. Rather than design new

algorithms that may generate a more involved analysis, we study the performance of the classical algorithms `online-gradient-descent` and `follow-the-perturbed-leader`, essentially unmodified, when the feedback is delayed. In the delayed setting, we prove that both algorithms have a simple regret bound of $O(\sqrt{D})$. These bounds collapse to match the well-known $O(\sqrt{T})$ regret bounds if there are no delays (i.e., where $D = T$).

**Paper organization**　In Section 2, we analyze the `online-gradient-descent` algorithm in the delayed setting, giving upper bounds on the regret as a function of the sum of delays $D$. In Section 3, we analyze the `follow-the-perturbed-leader` in the delayed setting and derive a regret bound in terms of $D$. Due to space constraints, extensions to `online-mirror-descent` and `follow-the-lazy-leader` are deferred to the appendix. We conclude and propose future directions in Section 4.

## 2　Delayed gradient descent

**Convex optimization**　In online convex optimization, the input domain $K$ is convex, and each cost function $f_t$ is convex. For this setting, Zinkevich proposed a simple online algorithm, called `online-gradient-descent`, designed as follows [1]. The first point, $x_1$, is picked in $K$ arbitrarily. After picking the $t$th point $x_t$, `online-gradient-descent` computes the gradient $\nabla f_t|_{x_t}$ of the loss function at $x_t$, and chooses $x_{t+1} = \pi_K(x_t - \eta \nabla f_t|_{x_t})$ in the subsequent round, for some parameter $\eta \in \mathbb{R}_{>0}$. Here, $\pi_K$ is the projection that maps a point $x'$ to its nearest point in $K$ (discussed further below). Zinkevich showed that, assuming the Euclidean diameter of $K$ and the Euclidean lengths of all gradients $\nabla f_t|_x$ are bounded by constants, `online-gradient-descent` has an optimal regret bound of $O(\sqrt{T})$.

**Delayed gradient descent**　In the delayed setting, the loss function $f_t$ is not necessarily given by the adversary before we pick the next point $x_{t+1}$ (or even at all). The natural generalization of `online-gradient-descent` to this setting is to process the convex loss functions and apply their gradients the moment they are delivered. That is, we update

$$x'_{t+1} = x_t - \eta \sum_{s \in \mathcal{F}_t} \nabla f_s|_{x_s},$$

for some fixed parameter $\eta$, and then project $x_{t+1} = \pi_K(x'_{t+1})$ back into $K$ to choose our $(t+1)$th point. In the setting of Zinkevich, we have $\mathcal{F}_t = \{t\}$ for each $t$, and this algorithm is exactly `online-gradient-descent`. Note that a gradient $\nabla f_s|_{x_s}$ does not need to be timestamped by the round $s$ from which it originates, which is required by the pooling strategies of Weinberger and Ordentlich [14] and Joulani *et al.* [15] in order to return the feedback to the appropriate learner.

**Theorem 2.1.** *Let $K$ be a convex set with diameter 1, let $f_1, \ldots, f_T$ be convex functions over $K$ with $\|\nabla f_t|_x\|_2 \leq L$ for all $x \in K$ and $t \in [T]$, and let $\eta \in \mathbb{R}$ be a fixed parameter. In the presence of adversarial delays,* `online-gradient-descent` *selects points $x_1, \ldots, x_T \in K$ such that for all $y \in K$,*

$$\sum_{t=1}^{T} f_t(x_t) - \sum_{t=1}^{T} f_t(y) = O\left(\frac{1}{\eta} + \eta L^2 (T + D)\right),$$

*where $D$ denotes the sum of delays over all rounds $t \in [T]$.*

For $\eta = 1/L\sqrt{T + D}$, Theorem 2.1 implies a regret bound of $O(L\sqrt{D + T}) = O(L\sqrt{D})$. This choice of $\eta$ requires prior knowledge of the final sum $D$. When this sum is not known, one can calculate $D$ on the fly: if there are $\delta$ outstanding (undelivered) cost functions at a round $t$, then $D$ increases by exactly $\delta$. Obviously, $\delta \leq T$ and $T \leq D$, so $D$ at most doubles. We can therefore employ the "doubling trick" of Auer *et al.* [26] to dynamically adjust $\eta$ as $D$ grows.

In the undelayed setting analyzed by Zinkevich, we have $D = T$, and the regret bound of Theorem 2.1 matches that obtained by Zinkevich. If each delay $d_t$ is bounded by some fixed value $\tau$, Theorem 2.1 implies a regret bound of $O(L\sqrt{\tau T})$ that matches that of Langford *et al.* [21]. In both of these special cases, the regret bound is known to be tight.

Before proving Theorem 2.1, we review basic definitions and facts on convexity. A function $f$ : $K \to \mathbb{R}$ is *convex* if

$$f((1-\alpha)x + \alpha y) \leq (1-\alpha)f(x) + \alpha f(y) \qquad \forall x, y \in K, \alpha \in [0,1].$$

If $f$ is differentiable, then $f$ is convex iff

$$f(x) + \nabla f|_x \cdot (y - x) \leq f(y) \qquad \forall x, y \in K. \tag{1}$$

For $f$ convex but not necessarily differentiable, a *subgradient* of $f$ at $x$ is any vector that can replace $\nabla f|_x$ in equation (1). The (possible empty) set of gradients of $f$ at $x$ is denoted by $\partial f(x)$.

The gradient descent may occasionally update along a gradient that takes us out of the constrained domain $K$. If $K$ is convex, then we can simply project the point back into $K$.

**Lemma 2.2.** *Let $K$ be a closed convex set in a normed linear space $X$ and $x \in X$ a point, and let $x' \in K$ be the closest point in $K$ to $x$. Then, for any point $y \in K$,*

$$\|x - y\|_2 \leq \|x' - y\|_2.$$

We let $\pi_K$ denote the map taking a point $x$ to its closest point in the convex set $K$.

*Proof of Theorem 2.1.* Let $y = \arg\min_{x \in K}(f_1(x) + \cdots + f_T(x))$ be the best point in hindsight at the end of all $T$ rounds. For $t \in [T]$, by convexity of $f_t$, we have,

$$f_t(y) \geq f_t(x_t) + \nabla f_t|_{x_t} \cdot (y - x_t).$$

Fix $t \in [T]$, and consider the distance between $x_{t+1}$ and $y$. By Lemma 2.2, we know that $\|x_{t+1} - y\|_2 \leq \|x'_{t+1} - y\|_2$, where $x'_{t+1} = x_t - \eta \sum_{s \in \mathcal{F}_t} \nabla f_s|_{x_s}$.

We split the sum of gradients applied in a single round and consider them one by one. For each $s \in \mathcal{F}_t$, let $\mathcal{F}_{t,s} = \{r \in \mathcal{F}_t : r < s\}$, and let $x_{t,s} = x_t - \eta \sum_{r \in \mathcal{F}_{t,s}} \nabla f_r|_{x_r}$. Suppose $\mathcal{F}_t$ is nonempty, and fix $s' = \max \mathcal{F}_t$ to be the last index in $\mathcal{F}_t$. By Lemma 2.2, we have,

$$\|x_{t+1} - y\|_2^2 \leq \|x'_{t+1} - y\|_2^2 = \|x_{t,s'} - \eta \nabla f_{s'}|_{x_{s'}} - y\|_2^2$$
$$= \|x_{t,s'} - y\|_2^2 - 2\eta\big(\nabla f_{s'}|_{x_{s'}} \cdot (x_{t,s'} - y)\big) + \eta^2 \|\nabla f_{s'}|_{x_{s'}}\|_2^2.$$

Repeatedly unrolling the first term in this fashion gives

$$\|x_{t+1} - y\|_2^2 \leq \|x_t - y\|_2^2 - 2\eta \sum_{s \in \mathcal{F}_t} \nabla f_s|_{x_s} \cdot (x_{t,s} - y) + \eta^2 \sum_{s \in \mathcal{F}_t} \|\nabla f_s|_{x_s}\|_2^2.$$

For each $s \in \mathcal{F}_t$, by convexity of $f$, we have,

$$-\nabla f_s|_{x_s} \cdot (x_{t,s} - y) = \nabla f_s|_{x_s} \cdot (y - x_{t,s}) = \nabla f_s|_{x_s} \cdot (y - x_s) + \nabla f_s|_{x_s} \cdot (x_s - x_{t,s})$$
$$\leq f_s(y) - f_s(x_s) + \nabla f_s|_{x_s} \cdot (x_s - x_{t,s}).$$

By assumption, we also have $\|\nabla f_s|_{x_s}\|_2 \leq L$ for each $s \in \mathcal{F}_t$. With respect to the distance between $x_{t+1}$ and $y$, this gives,

$$\|x_{t+1} - y\|_2^2 \leq \|x_t - y\|_2^2 + 2\eta \sum_{s \in \mathcal{F}_t} (f_s(y) - f_s(x_s) + \nabla f_s|_{x_s} \cdot (x_s - x_{t,s})) + \eta^2 \cdot |\mathcal{F}_t| \cdot L^2.$$

Solving this inequality for the regret terms $\sum_{s \in \mathcal{F}_t} f_s(x_s) - f_s(y)$ and taking the sum of inequalities over all rounds $t \in [T]$, we have,

$$\sum_{t=1}^T (f_t(x_t) - f_t(y)) = \sum_{t=1}^T \sum_{s \in \mathcal{F}_t} f_s(x_s) - f_s(y)$$

$$\leq \frac{1}{2\eta} \cdot \sum_{t=1}^T \left( \|x_t - y\|_2^2 - \|x_{t+1} - y\|_2^2 + 2\eta \sum_{s \in \mathcal{F}_t} \nabla f_s|_{x_s} \cdot (x_s - x_{t,s}) + \eta^2 \cdot |\mathcal{F}_t| \cdot L^2 \right)$$

$$= \frac{1}{2\eta} \left( \sum_{t=1}^T \|x_t - y\|_2^2 - \|x_{t+1} - y\|_2^2 \right) + \frac{\eta}{2} T L^2 + \sum_{t=1}^T \sum_{s \in \mathcal{F}_t} \nabla f_s|_{x_s} \cdot (x_s - x_{t,s})$$

$$\leq \frac{1}{2\eta} + \frac{\eta}{2} T L^2 + \sum_{t=1}^T \sum_{s \in \mathcal{F}_t} \nabla f_s|_{x_s} \cdot (x_s - x_{t,s}). \tag{2}$$

The first two terms are familiar from the standard analysis of `online-gradient-descent`. It remains to analyze the last sum, which we call the *delay term*.

Each summand $\nabla f_s|_{x_s} \cdot (x_s - x_{t,s})$ in the delay term contributes loss proportional to the distance between the point $x_s$ when the gradient $\nabla f_s|_{x_s}$ is generated and the point $x_{t,s}$ when the gradient is applied. This distance is created by the other gradients that are applied in between, and the number of such in-between gradients are intimately tied to the total delay, as follows. By Cauchy-Schwartz, the delay term is bounded above by

$$\sum_{t=1}^{T} \sum_{s \in \mathcal{F}_t} \nabla f_s|_{x_s} \cdot (x_s - x_{t,s}) \le \sum_{t=1}^{T} \sum_{s \in \mathcal{F}_t} \|\nabla f_s|_{x_s}\|_2 \|x_s - x_{t,s}\|_2 \le L \sum_{t=1}^{T} \sum_{s \in \mathcal{F}_t} \|x_s - x_{t,s}\|_2. \quad (3)$$

Consider a single term $\|x_s - x_{t,s}\|_2$ for fixed $t \in [T]$ and $s \in \mathcal{F}_t$. Intuitively, the difference $x_{t,s} - x_s$ is roughly the sum of gradients received between round $s$ and when we apply the gradient from round $s$ in round $t$. More precisely, by applying the triangle inequality and Lemma 2.2, we have,

$$\|x_{t,s} - x_s\|_2 \le \|x_{t,s} - x_t\|_2 + \|x_t - x_s\|_2 \le \|x_{t,s} - x_t\|_2 + \|x'_t - x_s\|_2.$$

For the same reason, we have $\|x'_t - x_s\|_2 \le \|x'_t - x_{t-1}\|_2 + \|x'_{t-1} - x_s\|_2$, and unrolling in this fashion, we have,

$$\|x_{t,s} - x_s\|_2 \le \|x_{t,s} - x_t\|_2 + \sum_{r=s}^{t-1} \|x'_{r+1} - x_r\|_2 \le \eta \sum_{p \in \mathcal{F}_{t,s}} \|\nabla f_p|_{x_p}\|_2 + \eta \sum_{r=s}^{t-1} \sum_{q \in \mathcal{F}_r} \|\nabla f_q|_{x_q}\|_2$$

$$\le \eta \cdot L \cdot \left( |\mathcal{F}_{t,s}| + \sum_{r=s}^{t-1} |\mathcal{F}_r| \right). \quad (4)$$

After substituting equation (4) into equation (3), it remains to bound the sum $\sum_{t=1}^{T} \sum_{s \in \mathcal{F}_t} (|\mathcal{F}_{t,s}| + \sum_{r=s}^{t-1} |\mathcal{F}_r|)$. Consider a single term $|\mathcal{F}_{t,s}| + \sum_{r=s}^{t-1} |\mathcal{F}_r|$ in the sum. This quantity counts, for a gradient $\nabla f_s|_{x_s}$ from round $s$ delivered just before round $t \ge s$, the number of other gradients that are applied while $\nabla f_s|_{x_s}$ is withheld. Fix two rounds $s$ and $t$, and consider an intermediate round $r \in \{s, \dots, t\}$. If $r < t$ then fix $q \in \mathcal{F}_r$, and if $r = t$ then fix $q \in \mathcal{F}_{t,s}$. The feedback from round $q$ is applied in a round $r$ between round $s$ and round $t$. We divide our analysis into two scenarios. In one case, $q \le s$, and the gradient from round $q$ appears only after $s$, as in the following diagram.

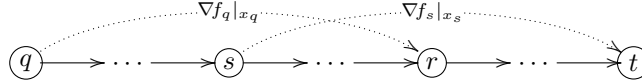

In the other case, $q > s$, as in the following diagram.

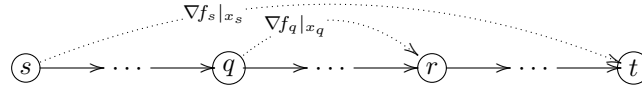

For each round $u$, let $d_u$ denote the number of rounds the gradient feedback is delayed (so $u \in \mathcal{F}_{u+d_u}$). There are at most $d_s$ instances of the latter case, since $q$ must lie in $s+1, \dots, t$. The first case can be charged to $d_q$. To bound the first case, observe that for fixed $q$, the number of indices $s$ such that $q < s \le d_q + q \le d_s + s$ is at most $d_q$. That is, all instances of the second case for a fixed $q$ can be charged to $d_q$. Between the two cases, we have $\sum_{t=1}^{T} \sum_{s \in \mathcal{F}_t} (|\mathcal{F}_{t,s}| + \sum_{r=s}^{t-1} |\mathcal{F}_r|) \le 2 \sum_{t=1}^{T} d_t$, and the delay term is bounded by

$$\sum_{t=1}^{T} \sum_{s \in \mathcal{F}_t} \nabla f_s|_{x_s} \cdot (x_s - x_{t,s}) \le 2\eta \cdot L^2 \sum_{t=1}^{T} d_t.$$

With respect to the overall regret, this gives,

$$\sum_{t=1}^{T} (f(x_t) - f(y)) \le \frac{1}{2\eta} + \eta \cdot L^2 \left( \frac{T}{2} + 2 \sum_{t=1}^{T} d_t \right) = O\left( \frac{1}{\eta} + \eta L^2 D \right),$$

as desired.
∎

*Remark* 2.3. The delay term $\sum_{t=1}^{T} \sum_{s \in \mathcal{F}_t} \nabla f_s|_{x_s} \cdot (x_s - x_{t,s})$ is a natural point of entry for a sharper analysis based on strong sparseness assumptions. The distance $x_s - x_{t,s}$ is measured by its projection against the gradient $\nabla f_s|_{x_s}$, and the preceding proof assumes the worst case and bounds the dot product with the Cauchy-Schwartz inequality. If, for example, we assume that gradients are pairwise orthogonal and analyze `online-gradient-descent` in the unconstrained setting, then the dot product $\nabla f_s|_{x_s} \cdot (x_s - x_{t,s})$ is 0 and the delay term vanishes altogether.

## 3 Delaying the Perturbed Leader

**Discrete online linear optimization** In discrete online linear optimization, the input domain $K \subset \mathbb{R}^n$ is a (possibly discrete) set with bounded diameter, and each cost function $f_t$ is of the form $f_t(x) = c_t \cdot x$ for a bounded-length cost vector $c_t$. The previous algorithm `online-gradient-descent` does not apply here because $K$ is not convex.

A natural algorithm for this problem is `follow-the-leader`. Each round $t$, let $y_t = \arg\min_{x \in K} x \cdot (c_1 + \cdots + c_t)$ be the optimum choice over the first $t$ cost vectors. The algorithm picking $y_t$ in round $t$ is called `be-the-leader`, and can be shown to have zero regret. Of course, `be-the-leader` is infeasible since the cost vector $c_t$ is revealed *after* picking $y_t$. `follow-the-leader` tries the next best thing, picking $y_{t-1}$ in round $t$. Unfortunately, this strategy can have linear regret, largely because it is a deterministic algorithm that can be manipulated by an adversary.

Kalai and Vempala [2] gave a simple and elegant correction called `follow-the-perturbed-leader`. Let $\epsilon > 0$ be a parameter to be fixed later, and let $Q_\epsilon = [0, 1/\epsilon]^n$ be the cube of length $1/\epsilon$. Each round $t$, `follow-the-perturbed-leader` randomly picks a vector $c_0 \in Q_\epsilon$ by the uniform distribution, and then selects $x_t = \arg\min_{x \in K} x \cdot (c_0 + \cdots + c_{t-1})$ to optimize over the previous costs plus the random perturbation $c_0$. With the diameter of $K$ and the lengths $\|c_t\|$ of each cost vector held constant, Kalai and Vempala showed that `follow-the-perturbed-leader` has regret $O(\sqrt{T})$ in expectation.

**Following the delayed and perturbed leader** More generally, `follow-the-perturbed-leader` optimizes over all information available to the algorithm, plus some additional noise to smoothen the worst-case analysis. If the cost vectors are delayed, we naturally interpret `follow-the-perturbed-leader` to optimize over all cost vectors $c_t$ delivered in time for round $t$ when picking its point $x_t$. That is, the $t$th leader becomes the best choice with respect to all cost vectors delivered in the first $t$ rounds:

$$y_t^{\mathrm{d}} = \arg\min_{x \in K} \sum_{s=1}^{t} \sum_{r \in \mathcal{F}_s} c_r \cdot x$$

(we use the superscript d to emphasize the delayed setting). The $t$th perturbed leader optimizes over all cost vectors delivered through the first $t$ rounds in addition to the random perturbation $c_0 \in Q_\epsilon$:

$$\tilde{y}_t^{\mathrm{d}} = \arg\min_{x \in K} \left( c_0 \cdot x + \sum_{s=1}^{t} \sum_{r \in \mathcal{F}_s} c_r \cdot x \right).$$

In the delayed setting, `follow-the-perturbed-leader` chooses $x_t = \tilde{y}_{t-1}^{\mathrm{d}}$ in round $t$. We claim that `follow-the-perturbed-leader` has a direct and simple regret bound in terms of the sum of delays $D$, that collapses to Kalai and Vempala's $O(\sqrt{T})$ regret bound in the undelayed setting.

**Theorem 3.1.** *Let $K \subseteq \mathbb{R}^n$ be a set with $L_1$-diameter $\leq 1$, $c_1, \ldots, c_T \in \mathbb{R}^n$ with $\|c_t\|_1 \leq 1$ for all $t$, and $\eta > 0$. In the presence of adversarial delays,* `follow-the-perturbed-leader` *picks points $x_1, \ldots, x_T \in K$ such that for all $y \in K$,*

$$\sum_{t=1}^{T} \mathbf{E}[c_t \cdot x_t] \leq \sum_{t=1}^{T} c_t \cdot y + O(\epsilon^{-1} + \epsilon D).$$

For $\epsilon = 1/\sqrt{D}$, Theorem 3.1 implies a regret bound of $O(\sqrt{D})$. When $D$ is not known *a priori*, the doubling trick can be used to adjust $\epsilon$ dynamically (see the discussion following Theorem 2.1).

To analyze `follow-the-perturbed-leader` in the presence of delays, we introduce the notion of a prophet, who is a sort of omniscient leader who sees the feedback immediately. Formally, the $t$th *prophet* is the best point with respect to all the cost vectors over the first $t$ rounds:

$$z_t = \underset{x \in K}{\arg\min}(c_1 + \cdots + c_t) \cdot x.$$

The $t$th *perturbed prophet* is the best point with respect to all the cost vectors over the first $t$ rounds, in addition to a perturbation $c_0 \in Q_\epsilon$:

$$\tilde{z}_t = \underset{x \in K}{\arg\min}(c_0 + c_1 + \cdots + c_t) \cdot x. \tag{5}$$

The prophets and perturbed prophets behave exactly as the leaders and perturbed leaders in the setting of Kalai and Vempala with no delays. In particular, we can apply the regret bound of Kalai and Vempala to the (infeasible) strategy of following the perturbed prophet.

**Lemma 3.2** ([2]). *Let $K \subseteq \mathbb{R}^n$ be a set with $L_1$-diameter $\leq 1$, let $c_1, \ldots, c_T \in \mathbb{R}^n$ be cost vectors bounded by $\|c_t\|_1 \leq 1$ for all $t$, and let $\epsilon > 0$. If $\tilde{z}_1, \ldots, \tilde{z}_{T-1} \in K$ are chosen per equation (5), then $\sum_{t=1}^{T} \mathbf{E}[c_t \cdot \tilde{z}_{i-1}] \leq \sum_{t=1}^{T} c_t \cdot y + O(\epsilon^{-1} + \epsilon T)$. for all $y \in K$.*

The analysis by Kalai and Vempala observes that when there are no delays, two consecutive perturbed leaders $\tilde{y}_t$ and $\tilde{y}_{t+1}$ are distributed similarly over the random noise [2, Lemma 3.2]. Instead, we will show that $\tilde{y}_t^{\mathrm{d}}$ and $\tilde{z}_t$ are distributed in proportion to delays. We first require a technical lemma that is implicit in [2].

**Lemma 3.3.** *Let $K$ be a set with $L_1$-diameter $\leq 1$, and let $u, v \in \mathbb{R}^n$ be vectors. Let $y, z \in \mathbb{R}^n$ be random vectors defined by $y = \arg\min_{y \in K}(q + u) \cdot y$ and $z = \arg\min_{z \in K}(q + v) \cdot z$, where $q$ is chosen uniformly at random from $Q = \prod_{i=1}^{n}[0, r]$, for some fixed length $r > 0$. Then, for any vector $c$,*

$$\mathbf{E}[c \cdot z] - \mathbf{E}[c \cdot y] \leq \frac{\|v - u\|_1 \|c\|_\infty}{r}.$$

*Proof.* Let $Q' = v + Q$ and $Q'' = u + Q$, and write $y = \arg\min_{y \in K} q'' \cdot y$ and $z = \arg\min_{z \in K} q' \cdot z$, where $q' \in Q'$ and $q'' \in Q''$ are chosen uniformly at random. Then

$$\mathbf{E}[c \cdot z] - \mathbf{E}[c \cdot y] = \mathbf{E}_{q'' \in Q''}[c \cdot z] - \mathbf{E}_{q' \in Q'}[c \cdot y].$$

Subtracting $\mathbf{P}[q' \in Q' \cap Q'']\mathbf{E}_{q' \in Q' \cap Q''}[c \cdot z]$ from both terms on the right, we have

$$\mathbf{E}_{q'' \in Q''}[c \cdot z] - \mathbf{E}_{q' \in Q'}[c \cdot y]$$
$$= \mathbf{P}[q'' \in Q'' \setminus Q'] \cdot \mathbf{E}_{q'' \in Q'' \setminus Q'}[c \cdot z] - \mathbf{P}[q' \in Q' \setminus Q''] \cdot \mathbf{E}_{q' \in Q' \setminus Q''}[c \cdot y]$$

By symmetry, $\mathbf{P}[q'' \in Q'' \setminus Q'] = \mathbf{P}[q' \in Q' \setminus Q'']$, and we have,

$$\mathbf{E}[c \cdot z] - \mathbf{E}[c \cdot y] \leq (\mathbf{P}[q'' \in Q'' \setminus Q'])\mathbf{E}_{q'' \in Q'' \setminus Q', q' \in Q' \setminus Q''}[c \cdot (z - y)].$$

By assumption, $K$ has $L_1$-diameter $\leq 1$, so $\|y - z\|_1 \leq 1$, and by Hölder's inequality, we have,

$$\mathbf{E}[c \cdot z] - \mathbf{E}[c \cdot y] \leq \mathbf{P}[q'' \in Q'' \setminus Q']\|c\|_\infty.$$

It remains to bound $\mathbf{P}[q'' \in Q'' \setminus Q'] = \mathbf{P}[q' \in Q' \setminus Q'']$. If $\|v - u\|_1 \leq r$, we have,

$$\mathrm{vol}(Q' \cap Q'') = \prod_{i=1}^{n}(r - |v_i - u_i|) = \mathrm{vol}(Q') \prod_{i=1}^{n}\left(1 - \frac{|(v_i - u_i)|}{r}\right) \geq \mathrm{vol}(Q')\left(1 - \frac{\|v - u\|_1}{r}\right).$$

Otherwise, if $\|u - v\|_1 > r$, then $\mathrm{vol}(Q' \cap Q'') = 0 \geq \mathrm{vol}(Q')(1 - \|v - u\|_1/r)$. In either case, we have,

$$\mathbf{P}[q' \in Q' \setminus Q''] = \frac{\mathrm{vol}(Q' \cap Q'')}{\mathrm{vol}(Q')} \leq 1 - \frac{\mathrm{vol}(Q' \cap Q'')}{\mathrm{vol}(Q')} \leq \frac{\|v - u\|_1}{r},$$

and the claim follows. ∎

Lemma 3.3 could also have been proven geometrically in similar fashion to Kalai and Vempala.

**Lemma 3.4.** $\sum_{t=1}^{T} \mathbf{E}[c_t \cdot \tilde{z}_{t-1}] - \mathbf{E}\left[c_t \cdot \tilde{y}_{t-1}^{\mathrm{d}}\right] \leq \epsilon D$, *where $D$ is the sum of delays of all cost vectors.*

*Proof.* Let $u_t = \sum_{s=1}^{t} c_t$ be the sum of all costs through the first $t$ rounds, and $v_t = \sum_{s:s+d_s \leq t} c_t$ be the sum of cost vectors actually delivered through the first $t$ rounds. Then the perturbed prophet $\tilde{z}_{t-1}$ optimizes over $c_0 + u_{t-1}$ and $\tilde{y}_{t-1}^{\mathrm{d}}$ optimizes over $c_0 + v_{t-1}$. By Lemma 3.3, for each $t$, we have

$$\mathbf{E}_{c_0 \sim Q_\epsilon}[c_t \cdot \tilde{z}_{t-1}] - \mathbf{E}_{c_0 \sim Q_\epsilon}\left[c_t \cdot \tilde{y}_{t-1}^{\mathrm{d}}\right] \leq \epsilon \cdot \|u_{t-1} - v_{t-1}\|_1 \|c_t\|_\infty \leq \epsilon \cdot |\{s < t : s + d_s \geq t\}|$$

Summed over all $T$ rounds, we have,

$$\sum_{t=1}^{T} \mathbf{E}_{c_0}[c_t \cdot \tilde{z}_t] - \mathbf{E}_{c_0}\left[c_t \cdot \tilde{y}_t^{\mathrm{d}}\right] \leq \epsilon \sum_{t=1}^{T} |\{s < t : s + d_s \geq t\}|.$$

The sum $\sum_{t=1}^{T} |\{s < t : s + d_s \geq t\}|$ charges each cost vector $c_s$ once for every round it is delayed, and therefore equals $D$. Thus, $\sum_{t=1}^{T} \mathbf{E}_{c_0}[c_t \cdot \tilde{z}_t] - \mathbf{E}_{c_0}\left[c_t \cdot \tilde{y}_t^{\mathrm{d}}\right] \leq \epsilon D$, as desired. ∎

Now we complete the proof of Theorem 3.1.

*Proof of Theorem 3.1.* By Lemma 3.4 and Lemma 3.2, we have,

$$\sum_{t=1}^{T} \mathbf{E}\left[c_t \cdot \tilde{y}_{t-1}^{\mathrm{d}}\right] \leq \sum_{t=1}^{T} \mathbf{E}[c_t \cdot \tilde{z}_{t-1}] + \epsilon D \leq \arg\min_{x \in K} \sum_{t=1}^{T} \mathbf{E}[c_t \cdot x] + O(\epsilon^{-1} + \epsilon D),$$

as desired. ∎

## 4   Conclusion

We prove $O(\sqrt{D})$ regret bounds for `online-gradient-descent` and `follow-the-perturbed-leader` in the delayed setting, directly extending the $O(\sqrt{T})$ regret bounds known in the undelayed setting. More importantly, by deriving a simple bound as a function of the delays, without any restriction on the delays, we establish a simple and intuitive model for measuring delayed learning. This work suggests natural relationships between the regret bounds of online learning algorithms and delays in the feedback.

Beyond analyzing existing algorithms, we hope that optimizing over the regret as a function of $D$ may inspire different (and hopefully simple) algorithms that readily model real world applications and scale nicely to distributed environments.

**Acknowledgements**   We thank Avrim Blum for introducing us to the area of online learning and helping us with several valuable discussions. We thank the reviewers for their careful and insightful reviews: finding errors, referencing relevant works, and suggesting a connection to mirror descent.

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
