[Supplementary Material · appendix.pdf]

# Online Learning with Adversarial Delays Appendix

Kent Quanrud[*] and Daniel Khashabi[†]

Department of Computer Science
University of Illinois at Urbana-Champaign
Urbana, IL 61801
{quanrud2,khashab2}@illinois.edu

## A  Mirror descent with delayed subgradients

**Geometries in $\mathbb{R}^n$ beyond $\|\cdot\|_2$**   The algorithm `online-gradient-descent`, presented in Section 2, is designed for convex loss functions $f_t$ whose subgradients $\nabla f_t|_x$ have bounded *Euclidean* length $\|\nabla f_t|_x\|_2$. While $\|\cdot\|_2$ is a geometrically intuitive norm to analyze, there are settings where other norms on $\mathbb{R}^n$ are more natural.

For example, in expert selection, we identify each dimension $i \in [n]$ with one of $n$ experts. Each round $t \in [T]$, we select an expert $i \in [n]$ and then receive a cost vector $c_t \in [-1, 1]^n$, where the $i$th coordinate $c_{t,i} \in [-1, 1]$ measures the loss incurred by expert $i$ for this round. More precisely, we choose a distribution of experts defined by a point $x_t$ in the probability simplex $K = \left\{ x \in [0,1]^n : \|x\|_1 \stackrel{\text{def}}{=} \sum_{i=1}^n |x_i| = 1 \right\}$. The $i$th coordinate $x_{t,i}$ corresponds to the probability of selecting expert $i$. The quantity $c_t \cdot x_t$ is the *expected* loss of the randomized strategy $x_t$. Note that the adversary selects $c_t$ with full knowledge of the randomized strategy $x_t$, but not the random bits that fix the choice of the $i$th expert.

A cost vector $c_t \in [-1, 1]^n$ can have Euclidean norm $\|c_t\|_2 = \sqrt{n}$, but its maximum norm is bounded by $\|c_t\|_\infty \stackrel{\text{def}}{=} \max\{|c_{t,1}|, \ldots, |c_{t,n}|\} \leq 1$. Because it operates in the Euclidean norm, `online-gradient-descent` obtains a regret bound on the order of $O(\sqrt{nT})$ in expert selection, and one would prefer a regret bound with respect to the $L_\infty$-bound of 1 rather than the Euclidean bound of $\sqrt{n}$. Indeed, a different algorithm that selects experts in proportion to the exponential of the costs (among others) improves the regret bounds from $O(\sqrt{nT})$ to $O(\sqrt{\ln(n)T})$ [1].

In this section, we analyze an algorithm called `online-mirror-descent`, introduced by Nemirovski and Yudin for (offline) convex minimization [2], that generalizes both `online-gradient-descent` and randomized expert selection by exponential weights[‡]. The algorithm has a diverse body of analyses and interpretations [4, 5, 6, 7, 8, 9, 10], and our presentation is particularly influenced by the survey [8] and the lecture notes [9].

**Additional results from convex analysis**   Before discussing `online-mirror-descent` in detail, we briefly review additional definitions and properties from convex analysis. These properties

---

[*] http://illinois.edu/~quanrud2/.  Supported in part by NSF grants CCF-1217462, CCF-1319376, CCF-1421231, CCF-1526799.

[†] http://illinois.edu/~khashab2/. Supported in part by a grant from Google.

[‡] Technically, the form of `online-mirror-descent` presented here generalizes a lazy version of `online-gradient-descent`, that applies a gradient $\nabla f_t|_{x'_t}$ to the unprojected point $x'_t$ rather than to the projected point $x_t = \pi_K(x'_t)$ to obtain the next unprojected point $x_t$. This variant of `online-gradient-descent` is also analyzed in [3].

are all well known. Our overview is not comprehensive and in particular we limit ourselves to $\mathbb{R}^n$ for ease of exposition. We refer the reader to [11, 12, 13, 14, 15] for further background.

Let $\varphi : \mathbb{R}^n \to \mathbb{R} \cup \{+\infty\}$ be a convex function. The *epigraph* of $\varphi$ is the set $\operatorname{epi} \varphi = \{(x, \mu) \in \mathbb{R}^n \times \mathbb{R} : \mu \geq \varphi(x)\}$ in the higher-dimensional space $\mathbb{R}^{n+1}$. The function $\varphi$ is *closed* if $\operatorname{epi} \varphi$ is a closed subset of $\mathbb{R}^n$. The closure of a function $\varphi$ is the closed function whose epigraph is the closure of $\operatorname{epi} \varphi$. If the function $\varphi$ is closed, then the closure of $\varphi$ is $\varphi$. The *domain* of $\varphi$ is the convex set of points $\operatorname{dom} \varphi \stackrel{\text{def}}{=} \{x \in \mathbb{R}^n : \varphi(x) < \infty\}$ where $\varphi$ is finite. The function $\varphi$ is *proper* if $\operatorname{dom} \varphi \neq \emptyset$.

Let $\varphi : \mathbb{R}^n \to \mathbb{R} \cup \{+\infty\}$ be a convex function. The *Fenchel conjugate* of $\varphi$ is the function $\bar{\varphi} : \mathbb{R}^n \to \mathbb{R}$ defined by $\bar{\varphi}(\bar{x}) = \sup_{x \in \mathbb{R}^n} \langle x, \bar{x} \rangle - \varphi(x)$. The conjugate function $\bar{\varphi}$ is a closed and convex function, and proper if and only if $\varphi$ is [11, Theorem 2.2]. The conjugate $\bar{\bar{\varphi}}$ of the conjugate function $\bar{\varphi}$ is the closure of the original function $\varphi$, and equals $\varphi$ if $\varphi$ is closed. The subgradients of $\varphi$ and $\bar{\varphi}$ are related by the following.

**Lemma A.1** ([11, Theorem 23.5]). *Let $\varphi : \mathbb{R}^n \to \mathbb{R} \cup \{+\infty\}$ be a closed and proper convex function, $x \in \mathbb{R}^n$, and $\bar{x} \in \mathbb{R}^n$. The following conditions are equivalent.*

*(i) $\bar{x} \in \partial \varphi(x)$.*
*(ii) The function $z \mapsto \langle z, \bar{x} \rangle - \varphi(z)$ attains its supremum in $z$ at $z = x$.*
*(iii) $\varphi(x) + \bar{\varphi}(\bar{x}) \leq \langle x, \bar{x} \rangle$.*
*(iv) $\varphi(x) + \bar{\varphi}(\bar{x}) = \langle x, \bar{x} \rangle$.*
*(v) $x \in \partial \bar{\varphi}(\bar{x})$.*
*(vi) The function $\bar{z} \mapsto \langle x, \bar{z} \rangle - \bar{\varphi}(\bar{z})$ attains its supremum at $\bar{z} = \bar{x}$.*

Consider any fixed norm $\|\cdot\|$ on $\mathbb{R}^n$. The *dual norm* of $\|\cdot\|$ is the norm $\|\cdot\|_\star$ on $\mathbb{R}^n$ defined by $\|y\|_\star = \sup_{\|x\|=1} |\langle x, y \rangle|$. A norm $\|\cdot\|$ and its dual norm $\|\cdot\|_\star$ satisfy a generalized form of the Cauchy-Schwartz inequality, $\langle x, y \rangle \leq \|x\| \|y\|_\star$. For example, the dual norm of the Euclidean norm $\|\cdot\|_2$ is again $\|\cdot\|_2$. The dual norm of the $L_\infty$-norm $\|\cdot\|_\infty$ is the $L_1$-norm $\|\cdot\|_1$. For $p > 1$, the dual norm of the $L_p$-norm $\|x\|_p \stackrel{\text{def}}{=} (\sum_{i=1}^n |x|^p)^{1/p}$ is the $L_q$-norm $\|\cdot\|_q$, where $q$ satisfies $1/p + 1/q = 1$.

Let $\varphi : \mathbb{R}^n \to \mathbb{R} \cup \{+\infty\}$ be a convex function. For a constant $\rho > 0$, we say $f$ is *smooth with modulus $\rho$* with respect to a norm $\|\cdot\|$ if it is differentiable and has $\rho$-Lipschitz-continuous gradients. That is, for any two points $x, y \in \mathbb{R}^n$, we have $\|\nabla \varphi|_x - \nabla \varphi|_y\|_\star \leq \rho \|x - y\|$. There are many equivalent conditions for a convex function to be smooth; see, for example, [15, Theorem 2.1.5] or [14, Section 3.5].

Let $\varphi : \mathbb{R}^n \to \mathbb{R} \cup \{+\infty\}$ be a convex function and $\sigma > 0$ a constant. The function $\varphi$ is *strongly convex with modulus $\sigma$* with respect to a norm $\|\cdot\|$ if the function $\psi(x) \stackrel{\text{def}}{=} \varphi(x) - (\sigma/2)\|x\|^2$ is also convex. As with smoothness, there are many equivalent properties for a function to be strongly convex; see, for example, [15, Section 2.1.3] or [14, Section 3.5]. In particular, for any two points $x, y \in \mathbb{R}^n$ and subgradient $\nabla \varphi|_x \in \partial \varphi(x)$, we have $\varphi(y) \geq \varphi(x) + \langle y - x, \nabla \varphi|_x \rangle + (\sigma/2)\|y - x\|^2$.

Smoothness and strong convexity are dual properties in the following sense.

**Lemma A.2** ([14, Section 3.5], [8, Lemma 2.19]). *Let $\varphi : \mathbb{R}^n \to \mathbb{R} \cup \{+\infty\}$ be a closed and convex function. Then $\varphi$ is strongly convex with modulus $\sigma$ with respect to a norm $\|\cdot\|$ if and only if $\bar{\varphi}$ is smooth with modulus $\sigma^{-1}$ with respect to the dual norm $\|\cdot\|_\star$.*

**Mirror descent**  Let $K$ be a closed convex set in $\mathbb{R}^n$. Fix a norm $\|\cdot\|$ in $\mathbb{R}^n$, and let $\varphi$ be a closed and strongly convex function on $\mathbb{R}^n$. The strongly convex function $\varphi$ is called the *regularizer*, and is required to have subgradients in the closed input domain $K$. Without loss of generality, we assume that $\varphi(x) = \infty$ over all nonfeasible points $x \notin K$.

Each round $t \in [T]$, `online-mirror-descent` has both a *primal point* $x_t \in K$ and a *dual point* $\bar{x}_t \in \mathbb{R}^n$. We choose the first dual point to be 0. Recall that by Lemma A.2, since $\varphi$ is strongly convex, its Fenchel conjugate $\bar{\varphi}$ is smooth and in particular differentiable everywhere. Given the $t$th dual point $\bar{x}_t$, the primal point $x_t$ is the derivative of the conjugate $\bar{\varphi}$ at $\bar{x}_t$, $x_t \stackrel{\text{def}}{=} \nabla \bar{\varphi}|_{\bar{x}_t}$. Given a subgradient $\nabla f_t|_{x_t} \in \partial f_t(x_t)$, we update the dual point against the gradient by the formula

$\bar{x}_{t+1} \stackrel{\text{def}}{=} \bar{x}_t - \eta \nabla f_t|_{x_t}$. The overall process looks like,

$$\ldots, \quad x_t \stackrel{\text{def}}{=} \nabla\bar{\varphi}|_{\bar{x}_t}, \quad \bar{x}_{t+1} \stackrel{\text{def}}{=} \bar{x}_t - \eta \nabla f_t|_{x_t}, \quad x_{t+1} \stackrel{\text{def}}{=} \nabla\bar{\varphi}|_{\bar{x}_{t+1}}, \quad \ldots$$

Intuitively, all the information (in the form of gradients) exists in the dual space (with norm $\|\cdot\|_\star$), and is "mirrored" back into the primal space (with the original norm $\|\cdot\|$) via the smooth conjugate function $\bar{\varphi}$.

If we inline the role of the dual point and apply Lemma A.1, the next point $x_{t+1}$ is chosen as a function of $x_t$ as

$$x_{t+1} = \nabla\bar{\varphi}|_{\bar{x}_{t+1}} = \nabla\bar{\varphi}|_{\bar{x}_t - \eta \nabla f_t|_{x_t}} = \arg\max_{x \in K}\{\langle x, \bar{x}_t - \eta \nabla f_t|_{x_t}\rangle - \varphi(x)\}.$$

If we expand this last term further, then the algorithm can be written equivalently as

$$\begin{aligned}
x_{t+1} &= \arg\max_{x \in K}\{\langle x, \bar{x}_t\rangle - \eta \langle x, \nabla f_t|_{x_t}\rangle - \varphi(x)\} \\
&= \arg\max_{x \in K}\{\varphi(x_t) + \langle x - x_t, \bar{x}_t\rangle - \eta \langle x, \nabla f_t|_{x_t}\rangle - \varphi(x)\} \\
&= \arg\max_{x \in K}\{-\eta \langle x, \nabla f_t|_{x_t}\rangle - D_\varphi(x|x_t, \bar{x}_t)\} \\
&= \arg\min_{x \in K}\{\langle x, \eta \nabla f_t|_{x_t}\rangle + D_\varphi(x|x_t, \bar{x}_t)\},
\end{aligned} \tag{1}$$

where $D_\varphi(x|x_t, \bar{x}_t) \stackrel{\text{def}}{=} \varphi(x) - \varphi(x_t) - \langle x - x_t, \bar{x}_t\rangle$ is the Bregman divergence of $\varphi$ from $x_t$ to $x$ with the choice of subgradient $\bar{x}_t \in \partial\varphi(x_t)$. The Bregman divergance $D_\varphi(x|x_t, \bar{x}_t)$ measures the amount of error between the actual value of $\varphi$ at $x$ and its first-order approximation from $x_t$ with respect to the subgradient $\bar{x}_t \in \partial\varphi(x_t)$.[§] By convexity of $\varphi$, $D_\varphi(\cdot|\cdot)$ is always nonnegative. Moreover, because $\varphi$ is strongly convex, $D_\varphi(x|x_t, \bar{x}_t)$ grows quadratically in the distance between $x$ and $x_t$. Despite these and other nice properties, $D_\varphi(x|x_t, \bar{x}_t)$ satisfies neither the triangle inequality nor symmetry. The final formulation of mirror descent in equation (1) chooses $x_{t+1}$ greedily against the direction of the gradient, but the greediness is tempered by the Bregman divergence $D_\varphi(\cdot|x_t, \bar{x}_t)$.

The Bregman divergence satisfies the following identity that we will use. For $x, y, z \in \mathbb{R}^n$, and subgradients $\bar{x} \in \partial\varphi(x)$ and $\bar{y} \in \partial\varphi(y)$, we have

$$\langle z - y, \bar{y} - \bar{x}\rangle = D_\varphi(z|x, \bar{x}) - D_\varphi(z|y, \bar{y}) - D_\varphi(y|x, \bar{x}) \tag{2}$$

(see, for example, [9, Fact 5.3.3]).

*Example* A.3. For a closed convex set $K$, consider the function

$$\varphi(x) \stackrel{\text{def}}{=} \begin{cases} \frac{1}{2}\|x\|_2^2 & \text{if } x \in K. \\ +\infty & \text{otherwise.} \end{cases}$$

The function $\varphi$ is strongly convex (with modulus 1) with respect to the Euclidean norm $\|\cdot\|_2$. The derivative of the conjugate $\bar{\varphi}$ takes the dual point $\bar{x} \in \mathbb{R}^n$ to its closest point in $K$. Thus `online-mirror-descent`, with this choice of norm and regularizer, resembles a lazy version of the `online-gradient-descent` algorithm considered in Section 2.

*Example* A.4 ([16, 4]). Let $K$ be the probability simplex in $\mathbb{R}^n$, and consider the negative entropy function

$$\varphi(x) \stackrel{\text{def}}{=} \begin{cases} \sum_{i=1}^n x_i \ln x_i & \text{if } x \in K, \\ +\infty & \text{otherwise} \end{cases}$$

where $0 \ln 0 \stackrel{\text{def}}{=} 0$. The negative entropy function $\varphi$ is strongly convex (with modulus 1) with respect to the $L_1$-norm, and its Frenchel conjugate is the log-partition function $\bar{\varphi}(\bar{x}) \stackrel{\text{def}}{=} \ln(\sum_{i=1}^n \exp(\bar{x}_i))$. The derivative of $\bar{\varphi}$ is defined coordinate-wise by $(\nabla\bar{\varphi}|_{\bar{x}})_i \stackrel{\text{def}}{=} \exp(\bar{x}_i)/(\sum_{i'=1}^n \exp(\bar{x}_{i'}))$. Note that $\nabla\bar{\varphi}|_{\bar{x}}$ is always a point in the probability simplex. In the context of expert selection, the `online-mirror-descent` algorithm with this choice of norm and regularizer randomly selects an expert $i$ in round $t$ in proportion to the exponential weight.

---

[§]We make the choice of subgradient explicit because $\varphi$ is not necessarily differentiable.

**Mirror descent in the delayed setting**    In the delayed setting, the subgradient of the loss function $f_t$ is not necessarily given by the adversary before we update the dual and pick the next point $x_{t+1}$ (or even at all). The natural generalization of `online-mirror-descent` to this setting is to process the convex loss functions and apply their subgradients to the dual point the moment they are delivered. That is, we update

$$\bar{x}_{t+1} = \bar{x}_t - \eta \sum_{s \in \mathcal{F}_t} \nabla f_s|_{x_s}$$

for some fixed parameter $\eta$, and then project $x_{t+1} = \nabla\bar{\varphi}|_{\bar{x}_{t+1}}$. In the undelayed setting, we have $\mathcal{F}_t = \{t\}$ for each $t$, and this algorithm is exactly `online-mirror-descent`.

**Theorem A.5.** *Let $K \subseteq \mathbb{R}^n$ be a closed convex set. For a fixed norm $\|\cdot\|$ on $\mathbb{R}^n$, let $\varphi : \mathbb{R}^n \to \mathbb{R} \cup \{+\infty\}$ be a closed and strongly convex function with modulus $\sigma$ and with $\mathrm{dom}\,\varphi = K$. Let $f_1, \ldots, f_T$ be convex functions with subgradients satisfying $\|\nabla f_t|_x\|_\star \le L$ for all $x \in K$ and $t \in T$. In the presence of adversarial delays, `online-mirror-descent` selects points $x_1, \ldots, x_T \in K$ such that for all $y \in K$,*

$$\sum_{t=1}^{T} f_t(x_t) - \sum_{t=1}^{T} f_t(y) = O\left(\frac{1}{\eta}(\varphi(y) - \varphi(x_1)) + \eta\frac{L^2(T+D)}{\sigma}\right).$$

In the gradient descent setup of example A.3, the term $\varphi(y) - \varphi(x_1)$ is bounded above by the square of the diameter $\Delta \stackrel{\text{def}}{=} \sup_{x,y \in K}\|x - y\|_2$ of $K$. For $\eta = \Theta(\Delta\sqrt{D}/L)$, we again achieve a regret bound on the order of $O(\Delta L\sqrt{D})$, matching Theorem 2.1 (there we treated $\Delta$ as a constant). In the expert selection setting of example A.4, $\varphi(y) - \varphi(x_1)$ is bounded above by $O(\ln(n))$, and for $\eta = \Theta(\sqrt{\ln(n)D})$, we achieve a regret bound on the order of $O(\sqrt{\ln(n)D})$.

*Proof of Theorem A.5.* Let $y \in K$ be any fixed point in $K$. By convexity, we first bound the regret by

$$\sum_{t=1}^{T}(f_t(x_t) - f_t(y)) \le \sum_{t=1}^{T} \langle x_t - y, \nabla f_t|_{x_t}\rangle. \tag{3}$$

The standard (undelayed) analysis for mirror descent proceeds to analyze each summand $\langle x_t - y, \nabla f_t|_{x_t}\rangle$ with the knowledge that $\nabla f_t|_{x_t}$ is applied to (the dual of) $x_t$. This is not true in the delayed setting, and we first align each gradient $\nabla f_s|_{x_s} \in \mathcal{F}_t$ alongside the primal point to which it is effectively applied.

As in the proof for `online-gradient-descent`, we split the sum of subgradients $\sum_{s \in \mathcal{F}_t} \nabla f_s|_{x_s}$ applied in a single round $t \in [T]$ and apply them one by one in increasing order of their originating round $s$. For each $s \in \mathcal{F}_t$, let $\mathcal{F}_{t,s} \stackrel{\text{def}}{=} \{r \in \mathcal{F}_t : r < s\}$ be the indices of other gradients that are applied in round $t$ before $\nabla f_s|_{x_s}$. Let $\bar{x}_s^- \stackrel{\text{def}}{=} \bar{x}_t - \eta \sum_{r \in \mathcal{F}_{t,s}} \nabla f_r|_{x_r}$ be the intermediate dual point to which we apply $\nabla f_s|_{x_s}$, and let $\bar{x}_s^+ \stackrel{\text{def}}{=} \bar{x}_s^- - \eta\nabla f_s|_{x_s}$ be the intermediate dual point after applying $\nabla f_s|_{x_s}$. Let $x_s^- \stackrel{\text{def}}{=} \nabla\bar{\varphi}|_{\bar{x}_s^-}$ and $x_s^+ \stackrel{\text{def}}{=} \nabla\bar{\varphi}|_{\bar{x}_s^+}$ be their projected primal points in $K$.

Rearranging the sum in equation (3) and organizing the gradients by point of delivery, we have

$$\sum_{t=1}^{T} \langle x_t - y, \nabla f_t|_{x_t}\rangle = \sum_{t=1}^{T} \sum_{s \in \mathcal{F}_t} \langle x_s - y, \nabla f_s|_{x_s}\rangle$$

$$= \sum_{t=1}^{T} \sum_{s \in \mathcal{F}_t} \langle x_s^- - y, \nabla f_s|_{x_s}\rangle + \sum_{t=1}^{T} \sum_{s \in \mathcal{F}_t} \langle x_s - x_s^-, \nabla f_s|_{x_s}\rangle. \tag{4}$$

The first sum is familiar from the standard analysis of undelayed mirror descent and will be analyzed by similar techniques. The second term generalizes the delay term in equation (2) from the earlier proof for `online-gradient-descent`.

To analyze the first sum, fix $t \in [T]$ and $s \in \mathcal{F}_t$, and expand the summand

$$\left\langle x_s^- - y, \nabla f_s|_{x_s} \right\rangle = \left\langle x_s^+ - y, \nabla f_s|_{x_s} \right\rangle + \left\langle x_s^- - x_s^+, \nabla f_s|_{x_s} \right\rangle.$$

The subsequent primal point $x_s^+$ minimizes the function $\psi_s(x) \stackrel{\text{def}}{=} \left\langle x, \eta \nabla f_s|_{x_s} - \bar{x}_s^- \right\rangle + \varphi(x)$ over $x \in K$. Since $\psi_s$ is convex, $K$ is convex, and $y \in K$, the first-order conditions at $x_s^+$ imply that $\left\langle y - x_s^+, \delta \right\rangle \geq 0$ for any subgradient $\delta$ of $\psi_s$ at $x_s^+$. In particular, at the subgradient $\eta \nabla f_s|_{x_s} - \bar{x}_s^- + \bar{x}_s^+ \in \partial \psi_s(x_s^+)$, we have,

$$\left\langle y - x_s^+, \eta \nabla f_s|_{x_s} - \bar{x}_s^- + \bar{x}_s^+ \right\rangle \geq 0;$$

rearranging, this implies that

$$\left\langle x_s^+ - y, \nabla f_s|_{x_s} \right\rangle \leq \frac{1}{\eta} \left\langle y - x_s^+, \bar{x}_s^+ - \bar{x}_s^- \right\rangle.$$

Furthermore, by the aforementioned identity in equation (2), we have

$$\left\langle y - x_s^+, \bar{x}_s^+ - \bar{x}_s^- \right\rangle = D_\varphi(y|x_s^-, \bar{x}_s^-) - D_\varphi(y|x_s^+, \bar{x}_s^+) - D_\varphi(x_s^+|x_s^-, \bar{x}_s^-).$$

Thus,

$$\begin{aligned}
\left\langle x_s^- - y, \nabla f_s|_{x_s} \right\rangle &= \left\langle x_s^+ - y, \nabla f_s|_{x_s} \right\rangle + \left\langle x_s^- - x_s^+, \nabla f_s|_{x_s} \right\rangle \\
&\leq \frac{1}{\eta} \left( D_\varphi(y|x_s^-, \bar{x}_s^-) - D_\varphi(y|x_s^+, \bar{x}_s^+) - D_\varphi(x_s^+|x_s^-, \bar{x}_s^-) \right) + \left\langle x_s^- - x_s^+, \nabla f_s|_{x_s} \right\rangle \\
&= \frac{1}{\eta} \left( D_\varphi(y|x_s^-, \bar{x}_s^-) - D_\varphi(y|x_s^+, \bar{x}_s^+) \right) + \frac{1}{\eta} \left( \left\langle x_s^- - x_s^+, \eta \nabla f_s|_{x_s} \right\rangle - D_\varphi(x_s^+|x_s^-, \bar{x}_s^-) \right). \quad (5)
\end{aligned}$$

The first term will later telescope over all rounds $s \in [T]$. Consider the second term $\left\langle x_s^- - x_s^+, \eta \nabla f_s|_{x_s} \right\rangle - D_\varphi(x_s^+|x_s^-, \bar{x}_s^-)$. Since $\varphi$ is strongly convex with modulus $\sigma$, we have $D_\varphi(x_s^+|x_s^-, \bar{x}_s^-) \geq \frac{\sigma}{2} \|x_s^+ - x_s^-\|^2$. Thus,

$$\begin{aligned}
\left\langle x_s^- - x_s^+, \eta \nabla f_s|_{x_s} \right\rangle &- D_\varphi(x_s^+|x_s^-, \bar{x}_s^-) \\
&\leq \left\langle x_s^- - x_s^+, \eta \nabla f_s|_{x_s} \right\rangle - \frac{\sigma}{2} \|x_s^+ - x_s^-\|^2 && \text{by strong convexity of } \varphi, \\
&\leq \|x_s^+ - x_s^-\| \|\eta \nabla f_s|_{x_s}\|_\star - \frac{\sigma}{2} \|x_s^+ - x_s^-\|^2 && \text{by Cauchy-Schwartz,} \\
&\leq \frac{1}{2\sigma} \|\eta \nabla f_s|_{x_s}\|_\star^2 && \text{by the identity } 2ab - a^2 \leq b^2, \\
&= O\left(\frac{\eta^2 L^2}{\sigma}\right) && \text{by assumption.} \quad (6)
\end{aligned}$$

Plugging equation (6) into equation (5), we have,

$$\left\langle x_s^- - y, \nabla f_s|_{x_s} \right\rangle \leq \frac{1}{\eta} \left( D_\varphi(y|x_s^-, \bar{x}_s^-) - D_\varphi(y|x_s^+, \bar{x}_s^+) \right) + O\left(\frac{\eta L^2}{\sigma}\right).$$

Summed over all $s \in \mathcal{F}_t$ and $t \in [T]$ and telescoping, we have,

$$\begin{aligned}
\sum_{t=1}^T \sum_{s \in \mathcal{F}_t} \left\langle x_s^- - y, \nabla f_s|_{x_s} \right\rangle &= \sum_{t=1}^T \sum_{s \in \mathcal{F}_t} \left( \frac{1}{\eta} \left( D_\varphi(y|x_s^-, \bar{x}_s^-) - D_\varphi(y|x_s^+, \bar{x}_s^+) \right) + O\left(\frac{\eta L^2}{\sigma}\right) \right) \\
&= \frac{1}{\eta} \left( D_\varphi(y|x_1, \bar{x}_1) - D_\varphi(y|x_{T+1}, \bar{x}_{T+1}) \right) + O\left(\frac{\eta L^2 T}{\sigma}\right). \quad (7)
\end{aligned}$$

arriving at the usual bound for (undelayed) mirror descent.

It remains to analyze the delay term $\sum_{t=1}^T \sum_{s \in \mathcal{F}_t} \left\langle x_s - x_s^-, \nabla f_s|_{x_s} \right\rangle$. As in the case of `online-gradient-descent`, each summand $\left\langle x_s - x_s^-, \nabla f_s|_{x_s} \right\rangle$ reflects the distance between the point $x_s$ when $\nabla f_s|_{x_s}$ is generated, and the point $x_s^-$ when $\nabla f_s|_{x_s}$ is applied. By Cauchy-Schwartz, the delay term is bounded above by

$$\sum_{t=1}^T \sum_{s \in \mathcal{F}_t} \left\langle x_s - x_s^-, \nabla f_s|_{x_s} \right\rangle \leq \sum_{t=1}^T \sum_{s \in \mathcal{F}_t} \|x_s - x_s^-\| \|\nabla f_s|_{x_s}\|_\star \leq L \sum_{t=1}^T \sum_{s \in \mathcal{F}_t} \|x_s - x_s^-\|.$$

Fix $s$ and consider the distance $\|x_s - x_s^-\|$. Applying the triangle inequality many times over, we first write

$$\left\|x_s - x_s^-\right\| \leq \sum_{r=s}^{t-1}\|x_r - x_{r+1}\| + \|x_t - x_{t,s}\|$$

Fix $r \in \{s, s+1, \ldots, t-1\}$ and consider a single term $\|x_r - x_{r+1}\|$. This is the distance traversed by the primal point from applying the gradients $\sum_{q \in \mathcal{F}_r} \nabla f_q|_{x_q}$ to $x_r$. In terms of derivatives of the dual regularizer, we have $\|x_r - x_{r+1}\| = \left\|\nabla\bar{\varphi}|_{\bar{x}_r} - \nabla\bar{\varphi}|_{\bar{x}_{r+1}}\right\|$, where $\bar{x}_{r+1} = \bar{x}_r - \eta\sum_{q \in \mathcal{F}_r}\nabla f_q|_{x_q}$. As the Fenchel conjugate of a strongly convex function with modulus $\sigma$, $\bar{\varphi}$ is smooth with modulus $\sigma^{-1}$, and in particular,

$$\left\|\nabla\bar{\varphi}|_{\bar{x}_r} - \nabla\bar{\varphi}|_{\bar{x}_{r+1}}\right\| = O\left(\frac{1}{\sigma}\|\bar{x}_r - \bar{x}_{r+1}\|_\star\right) = O\left(\frac{1}{\sigma}\left\|\eta\sum_{q \in \mathcal{F}_r}\nabla f_q|_{x_q}\right\|_\star\right).$$

Similarly, we have $\|x_t - x_{s,t}\| \leq O\left(\sigma^{-1}\left\|\eta\sum_{p \in \mathcal{F}_{t,s}}\nabla f_p|_{x_p}\right\|_\star\right)$. Thus,

$$
\begin{aligned}
\left\|x_s - x_s^-\right\| &= O\left(\sum_{r=s}^{t-1}\frac{1}{\sigma}\left\|\eta\sum_{q \in \mathcal{F}_r}\nabla f_q|_{x_q}\right\|_\star + \frac{1}{\sigma}\left\|\eta\sum_{p \in \mathcal{F}_{t,s}}\nabla f_p|_{x_p}\right\|_\star\right) \\
&= O\left(\frac{\eta}{\sigma}\left(\sum_{r=s}^{t-1}\sum_{q \in \mathcal{F}_r}\|\nabla f_q|_{x_q}\|_\star + \sum_{p \in \mathcal{F}_{t,s}}\|\nabla f_p|_{x_p}\|_\star\right)\right) \\
&= O\left(\frac{\eta L}{\sigma}\left(\sum_{r=s}^{t-1}|\mathcal{F}_r| + |\mathcal{F}_{t,s}|\right)\right).
\end{aligned}
$$

With respect to the delay term, we have

$$\sum_{t=1}^{T}\sum_{s \in \mathcal{F}_t}\left\langle x_s - x_s^-, \nabla f_s|_{x_s}\right\rangle \leq L\sum_{t=1}^{T}\sum_{s \in \mathcal{F}_s}\left\|x_s^- - x_s\right\| = O\left(\frac{\eta L^2}{\sigma}\sum_{t=1}^{T}\sum_{s \in \mathcal{F}_s}\left(\sum_{r=s}^{t-1}|\mathcal{F}_r| + |\mathcal{F}_{t,s}|\right)\right).$$

This sum was encountered earlier in the proof for `online-gradient-descent`, and the same argument shows that the delay term is bounded by $\sum_{t=1}^{T}\sum_{s \in \mathcal{F}_t}\langle x_s - x_s^-, \nabla f_s|_{x_s}\rangle = O(\eta L^2 D/\sigma)$. Plugging this inequality along with equation (7) into equation (3), we bound the regret by

$$
\begin{aligned}
R(T) &\leq \frac{1}{\eta}\left(D_\varphi(y|x_1, \bar{x}_1) - D_\varphi(y|x_{T+1}, \bar{x}_{T+1})\right) + \eta\frac{TL^2}{2\sigma} + O\left(\eta\frac{DL^2}{\sigma}\right) \\
&= O\left(\frac{1}{\eta}(\varphi(y) - \varphi(x_1)) + \eta\frac{L^2(T+D)}{\sigma}\right),
\end{aligned}
$$

as desired. ∎

## B  Late and lazy leaders

A basic application of online combinatorial optimization is adaptive data structures. For example, in the *tree update problem*, one maintains a binary search tree over a collection of $n$ items, and serves an unknown sequence of access queries. If the cost of accessing an item is proportional to the depth of the item in the tree, then one prefers to have frequently placed items near the root, and less frequent items at the leaves. One classical data structure for this problem is the splay tree, by Sleater and Tarjan, who also introduced the tree update problem [17]. Kalai and Vempala observe that this problem can partially be solved by `follow-the-perturbed-leader`. We simply track the sum of access costs for each item, and each round we rebuild an optimal binary search tree with respect to those costs (plus a perturbation) by dynamic programming in $O(n^2)$ time. Compared to the cost of the best fixed tree in hindsight, this algorithm has an additive loss of $O(n\sqrt{nT})$ to serve

$T$ requests. This bound overlooks the fact that rebuilding a binary search tree between every two queries is absurdly inefficient.

To model this and similar problems efficiently, one introduces the notion of a *switching cost*, which is a fixed value that is charged every time we change our decision between subsequent rounds. For this setting, Kalai and Vempala gave a similar algorithm called `follow-the-lazy-leader`, which is nearly equivalent to `follow-the-perturbed-leader`, but changes decisions very infrequently. Happily, `follow-the-lazy-leader` is just as simple as `follow-the-perturbed-leader`.

Initially, the algorithm picks a point $p \in Q_\epsilon = \left[0, \epsilon^{-1}\right]^n$ uniformly at random. This defines a grid, $G_\epsilon = \left\{p + \epsilon^{-1}z : z \in \mathbb{Z}^n\right\}$, of width $\epsilon^{-1}$ and shifted randomly. Rather than follow the perturbed leader, we follow the leader and round it up to its nearest grid point. Formally, we define the $t$th *lazy leader* to be the unique grid point $\overline{y}_t \in G_\epsilon$ of the square $y_t + Q_0$. Because the grid is randomly shifted by the same distribution as the perturbation in `follow-the-perturbed-leader`, the perturbed leader and lazy leader are distributed identically and have identical expected costs [18, Lemma 1.2]. This observation immediately implies that `follow-the-lazy-leader` obtains regret $O(\sqrt{T})$[¶].

Extending `follow-the-lazy-leader` to the delayed setting has natural applications. For example, in the tree update problem, it is preferable to be able to serve many queries simultaneously, and to keep serving requests from an outdated search tree while building the next tree in parallel. One simple implementation to satisfy these requirement is to keep a pointer to the tree in *software transactional memory* (STM) [19]. When `follow-the-lazy-leader` shifts grid points, we build the new search tree in the background and swap the pointer in the STM once the new tree is completed. Here, the delays arise from serving requests from an outdated tree before the tree is replaced by a new one.

**Theorem B.1.** *Let $K \subseteq \mathbb{R}^n$ be a set with $L_1$-diameter $\leq 1$, let $c_1, \ldots, c_T \in \mathbb{R}^n_{\geq 0}$ be non-negative vectors with $\|c_t\|_1 \leq 1$ for all $t$, and let $\epsilon > 0$. In the presence of adversarial delays,* `follow-the-lazy-leader` *picks points $x_1, \ldots, x_T \in K$ such that two consecutive points $x_i$ and $x_{i+1}$ differ at most $\epsilon T$ times, and for all $y \in K$,*

$$\sum_{t=1}^{T} \mathbf{E}[c_t \cdot x_t] \leq \sum_{t=1}^{T} c_t \cdot y + O\big(\epsilon^{-1} + \epsilon D\big).$$

*Proof.* In the delayed setting, the $t$th delayed leader $\overline{y}_t^{\mathrm{d}}$ is the best point with respect to the grid point corresponding to the sum of all feedback through the first $t$ rounds. More formally, let $g : \mathbb{R}^n \to G_\epsilon$ be the mapping each point to the first grid point greater than or equal in each dimension to the point. Formally, we have

$$\overline{y}_t^{\mathrm{d}} = \arg\min_{x \in K} x \cdot g\left(\sum_{s=1}^{t} \sum_{r \in \mathcal{F}_s} c_r\right)$$

for each round $t$ (as before, we use the superscript $\mathrm{d}$ to emphasize the delayed setting). Following the lazy leader is the strategy that chooses the $t$th point $x_t$ as $x_t = \overline{y}_{t-1}^{\mathrm{d}}$ for each round $t$. Observe that since each cost vector $c_i$ is nonnegative, and their sum has $L_1$-norm $\|c_1 + \cdots + c_T\|_1 \leq T$, $\overline{y}_t^{\mathrm{d}}$ changes grid points at most $\epsilon T$ times.

The same proof as [18, Lemma 1.2] shows that each $\overline{y}_t^{\mathrm{d}}$ is distributed identically as $\tilde{y}_t^{\mathrm{d}}$. By linearity of expectation, $\mathbf{E}\big[c_t \cdot \overline{y}_{t-1}^{\mathrm{d}}\big] = \mathbf{E}\big[c_t \cdot \tilde{y}_{t-1}^{\mathrm{d}}\big]$ for all $t \in [T]$. Summing this equality over all rounds $t \in [T]$ and applying the regret bound for `follow-the-perturbed-leader` (Theorem 3.1) gives the bounds we seek. ∎