[Reviews · NeurIPS 2015]

Submitted by Assigned_Reviewer_1

Summary: This paper contributes to a growing literature which seeks to weaken

full-feedback assumptions in traditional online learning.

Here, rather than observing feedback immediately, the learner may have to wait an unknown amount of time (the "delay") before observing the results of actions taken on a given round.

The paper derives algorithms for this setting, where the delays are chosen adversarially -- more precisely, the paper shows that in fact existing learning algorithms already offer regret guarantees for this setting with appropriate choices of the learning rate.

Results for online gradient descent, follow the perturbed leader, and follow the lazy leader are given.

Praise: The results are intriguing, as they show once again that the standard online learning algorithms are quite robust to changes in the model.

The proofs are reasonably concise and to the point.

Criticism: The first question I had, given that "adversarial delays" appears in the title, was: how does the adversary behave here?

I was disappointed to see that there was no such discussion on this point.

In particular, are these bounds tight?

The discussion of Langford et al. before the proof of Thm 1 suggests that the optimal delays for the adversary, given a budget of D, is to spread them out evenly, and this should yield a tight bound.

But what does adversarial mean if there is no budget D, which seems to be the setting in question?

Perhaps "unknown delays" would be a better term here.

My second criticism is the lack of intuition given about the results.

In particular, it seems to me that the bounds are saying that additional delays are "equivalent" to playing more rounds of the game -- I would have liked to see some discussion about why this might be the case.

Is the intuition that delays are cumulative in the above sense grounded in the proof?

The proofs are reasonably clean yet obscure this point (to me).

In my opinion at least one of the proofs should move to the appendix to make way for a thorough discussion on the nature of the bounds.

Comments: page 1 (the big-O notation assumes that diameter of K): Typo page 2 (et al): Here and elsewhere, should be "et al." page 2 (for some parameter   R): Positive as well page 6 (For = O(1/  D), Theorem 2 implies a regret bound of O(  D).): Remove first O(), otherwise the statement is false. page 8 (regularized-follow-the-leader): Usually this is follow-the-regularized-leader
Summary: The results are nice but not groundbreaking.

Needs more intuition and consideration of the adversary.

Submitted by Assigned_Reviewer_2

The authors slightly modify two well known online algorithms, OGD and FTPL, to address the problem of adversarial delays. In this problem, it is assumed that adversary might not reveal the loss functions at end of every round. The modifications to the algorithms are most natural and I appreciate the simplicity.

Since the crux of the paper lies in the analysis of the modified algorithms, I need to be sure about the correctness of the proofs before accepting the paper. While it is of course possible that I have misunderstood a couple of things, I want to be convinced that it is my mistake rather than mistake in the proofs before agreeing that the paper can be accepted to a conference of such prestige.

Originality- Unless someone has already published the ideas before, I cannot question the originality of the work.

Significance- The authors give a few examples in the beginning as to why adversarial feedback might be delayed. I have not encountered this problem before; so I am assuming that this problem is of interest to the online learning community.

Clarity- The introduction and the two (modified) algorithms are certainly clear and simple to understand. However, I have a few issues regarding the presentation of the proofs. Normally, it would not matter, but considering the paper is all about the analysis, and the proofs are in the main paper, I think the presentation is important.

1. Line 033- Why is the game fundamentally unfair? Unfair to whom? Learner or adversary? If the adversary is adaptive, I would say that the adversary is very powerful. Moreover, considering the regret is w.r.t to the best single decision learner could have taken in hindsight, I do not understand the unfairness issue. 2. It is confusing how the proof is going from line 151 to 156. While the inequality in line 156 is correct ( I checked it- simply write ||x_{t+1}- y||^2= ||x_t - \sum_{s \in F_t} \nabla f_s|x_s - y||^2 and open up the expression on the right followed by simple algebra), I find what is written in the paper confusing. First, the authors fix s as max F_t, but then the expression is unrolled over $s$. Is s fixed or variable? 3. Please check the expression for x_{t,s} when it is introduced. I think in the gradient term, it should be \nabla f_r|x_r, not \nabla f_s|x_r.

4. I found it very difficult to understand the proof of Lemma 3.2. Can it be presented in a better way? It is simply the end part of Theorem 1.1 and Lemma 3.2 in Kalai-Vempala FTPL paper. In their paper, it is so simple to understand (the argument of what fraction of two hypercubes overlap followed by the fact that in the fraction they dont overlap, the maximum difference in decision*cost factor is bounded by L_1 norm of decision and L_{infty} norm of cost, which is precisely what is being done in Lemma 3.2 of this paper). I dont see the need to recast Kalai-Vempala's argument in a more complicated way. 5. There are number of notational mistakes in Lemma 3.3. Please check.

Quality- The main questions I have are in the proof of Theorem 1. 1. Why does the inequality in Line 192 hold? Specifically, why is x_{t,s} closer to x'_{s+1} in Euclidean space than to x_{s+1}? I simply could not understand how this came about? As far as as I can see, x_{t,s} is an arbitrarily defined point. Is x_{t,s} inside or outside the convex set K? Either way, why does the inequality hold? 2. I simply could not wrap my head around the argument between Line 200 to line 219. I tried to reason independently but could not. While of course it does not mean the proof is wrong, I would like to talk to other reviewers before being convinced about it.

Summary: I liked the paper and was generally happy with the fact that authors maintained simplicity, analyzing (most natural) modifications to two well known online algorithms, OGD and FTPL. The main reason for not accepting the paper is a doubt I have about the proof of Thm.1,

and couple of relatively minor issues which I highlight below. I will surely reconsider after author feedback and discussions.

Submitted by Assigned_Reviewer_3

The paper uses the on-line optimisation framework.

Suppose that the feedback (the cost function f_t) comes to us with arbitrary (possibly adversarial) delays. What bounds can we achieve?

The paper takes two algorithms, on-line gradient descent and follow the perturbed leader (in the optimisation context) and makes natural adjustments to them: all modifications to the x_t and to the current choice of the leader are made on the basis of the feedback available so far. New modifications are introduced as soon as feedback becomes available.

The paper provides theoretical bounds for the new algorithms. The change amounts to the introduction of an extra term of the order square root of the sum of delays, which is comparable with the standard \sqrt{T}.

The paper also considers an application to the tree update problem, but the discussion is rather terse.

The paper is lucidly written.

I believe the paper provides an important contribution to the theory of on-line prediction.

Minor issues:

1. Page 5, last displayed equation. I think brackets are needed after argmin for clarity.

2. I would like to point out some previous work concerned with the delays in feedback. See the treatment of weak and lazy teachers in

D. Ryabko. On the Flexibility of Theoretical Models for Pattern Recognition, Ph.D. Thesis; Royal Holloway, University of London. April, 2005
Summary: The paper adapts two on-line optimisation algorithms for the scenario where there is delay in the feedback.

Author Feedback
Author rebuttal: We thank all the reviewers for their careful reviews. We have generally incorporated all the changes suggested, addressed individually below. These changes have improved the paper and we are grateful for the feedback. Due to space constraints, not all comments can be addressed here. We omit some points that we agree with and have made simple adjustments for.

Assigned_Review_1 (AR1): Thank you for the additional references. They have both been incorporated into the paper. We have given considerable space to your first reference, "Online learning under delayed feedback", explained below in the section addressed to AR5.

AR2: Thank you for your careful review. The term "adversarial" was to imply no assumptions regarding bounded or independent delays, in the some way that we have "adversarial" or "worst-case" inputs in classical algorithms. A better term here may be "arbitrary" or "unbounded", as well as "unknown" as you've suggested. We've slowed the pace and softened our language when introducing the model accordingly. Regarding intuition, our hope was to obtain results clean and transparent enough to wrap one's mind around and extend. At your suggestion, we've tried to improve the commentary surrounding the results.

AR3: Thank you for your encouraging review. Thank you also for the reference to Ryabko's thesis, which is now cited.

AR4: Thank you for your astute review.

(Clarity) (1) We apologize for the confusion caused by our choice of words. The goal here is to gently motivate and introduce the notion of regret to an outside reader. We have softened the language and emphasized that the comment is informal.

(4) Our proof of Lemma 3.2 merely extends the argument of Kalai and Vempala. As authors, we went back and forth between the algebraic argument and the geometric one. In the end, we resolved that the algebraic argument may be more concrete and easy to verify, at the cost of Kalai and Vempala's (very nice) geometric intuition. As per your suggestion, we have provided comments that relate our algebraic version of the proof to the original geometric version.

(Quality) (1) You are correct. We should have written the proof to unroll from the recent rounds (t,t-1,t-2,...) to the oldest rather than the other way around (s,s+1,s+2,...). For example, the first step is use the triangle inequality and Lemma A.1 to obtain

||x_{t,s} - x_s}|| \leq ||x_{t,s} - x_t|| + ||x_t - x_s|| \leq ||x_{t,s} - x_t|| + ||x_t' - x_s||

The second term ||x_t' - x_s|| is similarly bounded by ||x_t' - x_{t-1}|| + ||x_{t-1}' - x_s||, and so on, until arriving at the same sum, and continuing the proof as before.

We are grateful to the reviewer for noting this error, and have corrected our proof accordingly.

AR5: Thank you for the reference to the paper of Joulani etal, also suggested by AR1. We have now included an introduction to this work and other work (such as that of Weinberger and Ordentlich) similar in spirit. As a different approach to a similar problem, these results provide a healthy contrast to our work. Please note the following.

(a) The framework of using a pool of learners, each of which acts as an undelayed learner over a subsequence of rounds, is a qualitatively different algorithm. Maintaining independent instances of learners proportional to the delay is more resource intensive then a single learning algorithm. Furthermore, the pooling framework requires the feedback to be timestamped by its originating round in order to return the feedback to the appropriate learner.

(b) The $\sqrt{D}$ bound is asymptotically sharper then the bounds obtained by the pooling framework. To recap, let $C$ be the maximum number of outstanding feedbacks at any single point of decision. Then the pooling framework has a regret bound of $O(\sqrt{CT})$. (Jensen's inequality can then model $C$ probabilistically.) Clearly, $D \leq CT$ in any scenario. For a simple example to distinguish the two bounds, suppose that the feedback from the first $\sqrt{T}$ rounds is withheld until after round $\sqrt{T} + 1$, and all other feedback is delivered on time. Then the maximum number of outstanding feedbacks is $C = \sqrt{T}$, giving a regret bound on the order of $T^{3/4}$. On the other hand, the total delay is still $D = O(T)$, and our results give a regret bound on the order of $\sqrt{T}$.

AR6: Thank you for your insightful observation. We think it may be correct. As noted in the conclusion, we expect a $\sqrt{D}$ regret bound for mirror descent, and similar bounds for most popular algorithms in online learning. The staying power of this work is the introduction of the quantity $D$, by which one can argue that standard algorithms have been fairly robust to delays all along. The individual proofs for gradient-descent and follow-the-perturbed-leader are experiments that validate the model. We look forward to updating the conclusion once we finalize our verification, very soon.